# Neutrophil to Lymphocyte Ratio (NLR) as an Easily Accessible Parameter for Monitoring Tacrolimus Overdose after Heart Transplantation—Experimental Study

**DOI:** 10.3390/diagnostics12010037

**Published:** 2021-12-24

**Authors:** Tomasz Urbanowicz, Anna Olasińska-Wiśniewska, Michał Michalak, Michał Rodzki, Anna Witkowska, Ewa Straburzyńska-Migaj, Bartłomiej Perek, Marek Jemielity

**Affiliations:** 1Cardiac Surgery and Transplantology Department, Poznan University of Medical Sciences, 61-848 Poznan, Poland; anna.olasinska@poczta.onet.pl (A.O.-W.); michal.rodzki@skpp.edu.pl (M.R.); anna.witkowska@skpp.edu.pl (A.W.); bperek@ump.edu.pl (B.P.); mjemielity@poczta.onet.pl (M.J.); 2Department of Computer Science and Statistics, Poznan University of Medical Sciences, 61-848 Poznan, Poland; michal@ump.edu.pl; 31st Department of Cardiology, Poznan University of Medical Sciences, 61-848 Poznan, Poland; ewa.straburzynska-migaj@skpp.edu.pl

**Keywords:** heart transplantation, tacrolimus, NLR, MPV, drug overdose

## Abstract

(1) Background: The combination of candidate selection, immunosuppressive therapy adjustment, and scrutinous monitoring is a cornerstone for optimizing long-term survival after a heart transplant. Neutrophil-to-lymphocyte ratio (NLR) is a simple marker of inflammatory reactions activation and may play a clinical role as a predictive marker in oncological and cardiovascular diseases. The aim of the study was to find simple hematologic indices available from whole blood count to help in immunosuppressive therapy monitoring. (2) Methods: Thirty patients (23 men and 7 women) with a mean age of 43 +/− 13 years who had undergone orthotopic heart transplantation were enrolled into the study. The blood samples for whole blood count and Tacrolimus level were collected during outpatient visits in heart transplant recipients every two months for 18 months after first year post transplantation. (3) Results: There was a significant correlation between Tacrolimus overdose (>15 ng/mL) and NLR (Spearman’s rho 0.99, *p* < 0.001) and mean platelet volume (Spearman’s rho 0.989, *p* < 0.001). The ROC analysis for NLR above 3.62 showed predictive properties for Tacrolimus overdose (over 15 ng/mL) (AUC =0.633, *p* = 0.008) with a sensitivity of 50% and specificity of 83.73%. (4) Conclusions: Our results suggest that NLR above 3.62 may be regarded as a simple indicator of tacrolimus overdose.

## 1. Introduction

Advanced heart failure involves symptoms affecting daily life activity, followed by high mortality, and it remains a major clinical and public health problem [1]. These patients may be treated, ultimately, with heart transplantation, characterized by 1-year survival approaching 80 to 90% [2]. The combination of candidate selection, immunosuppressive therapy adjustment and scrutinous monitoring is a cornerstone for optimizing long-term survival. Contemporary therapeutical protocols are based on calcineurin inhibitors [3].

A review performed by Hussain et al. presented a wide variety of problems influencing adherence to immunosuppression in transplant patients and concluded that there were modifiable factors to address to create improvements [4].

We believe that incorporating simple laboratory markers indicating the serum level disturbances in patients taking immunosuppressive drugs should be regarded as of primary value. Notably, drug level monitoring is possible only in reference hospitals. Therefore, the aim of the study was to find simple hematologic indices available from whole blood count to help in immunosuppressive therapy monitoring—even in outpatient clinics without specialistic laboratory support.

Neutrophil to lymphocyte ratio (NLR) is a simple marker of inflammatory reactions activation and has been postulated to play a clinical role as a predictive marker in oncology and cardiology [5,6,7,8,9]. The relationship between low values of NLR and rejection episodes in corneal transplantation has already been presented [10]. Park et al. presented evidence of correlation between NLR and overall and graft survival after liver transplantation [11].

Mean platelet volume (MPV) is another parameter representing induction of inflammatory reaction in different clinical situations [12,13,14]. Platelet indices were linked with worse cardiac function and adverse clinical outcomes in patients with heart failure in the MyoVasc study [15].

The aim of the study was to compare NLR and additionally MPV to tacrolimus serum levels as possible predictors for immunosuppression overdose after heart transplantation.

## 2. Material and Method

We evaluated 30 patients (23 men and 7 women) with a mean age of 43 ± 13 years who had undergone orthotropic heart transplantation between 2014–2019 in our department. The demographical and clinical data are presented in Table 1.

Following the transplantation, the patients were on immunosuppressive therapy, including following protocols of triple therapy with tacrolimus, mycophenolate mophetil, and steroids in 27 (90%) patients and double therapy with tacrolimus and steroids in 2 (10%). The mycophenolate mophetil was withdrawn in 2 cases due to gastrointestinal adverse reactions.

## 3. Method

The study period beginning was established as the 12th month after heart transplant and, subsequently, patients were followed up with for 18 months until the 30th month after transplant. The blood samples were collected during each outpatient visit between 12th to 30th month post transplantation. NLR and MPV indices were evaluated from whole blood count, as presented in Table 2. Similarly, the tacrolimus C-0 serum levels were measured during outpatient controls performed within 2-month intervals during a total of 18 months of observation (Table 2). The tacrolimus serum C-0 dose recommended for the analyzed treatment period was within the range of 5–15 ng/mL.

Patients with concomitant infection were excluded from the study.

## 4. Results

The separate measurements (18 months of time) were collected, totaling 300 tests in total. The tacrolimus (FK506) and mycophenolate mophetil serum levels were collected as C-0 levels (serum concentration before morning dose of drug administration). During the study period, patients were on low doses of steroid therapy, including 5 +/− 2.5 mg of prednisolone daily. Mycophenolate mophetil serum C-0 levels were estimated as 2.9 +/− 0.3 mcg/mL.

There were neither episodes of death nor rejection during the analyzed study period.

The NLR and MPV values were calculated from each whoule blood count. Then, both indices were correlated with Tacrolimus C-0 serum levels. The cutoff point for Tacrolimus was 15 ng/mL as upper threshold for C-0 concentration. There was one sample with a value of 15 ng/mL that was excluded from the analysis. There were 48 positive samples for Tacr concentration above 15 ng/mL and 241 more representing the Tacr serum levels below the cutoff point. The median values of Tacr were 17.6 (16.1–20.1) vs. 9 (7.5–11.3) (*p* < 0.01), respectively. The NLR values for Tacr above and below 15 ng/mL were 3.62 (2.12–16.83) vs. 2.41 (1.81–3.75) (*p* = 0.001), respectively. The MPV values for Tacr above and below 15 ng/mL were 8.1 (7.7–9.6) vs. 8.4 (7.6–9.2), *p* = < 0.01, respectively.

There was a significant correlation between Tacrolimus overdose (>15 ng/mL), NLR (Spearman’s rho 0.99, *p* < 0.001), and MPV (Spearman’s rho 0.989, *p* < 0.001) values, as presented in Figure 1 and Figure 2.

We also performed receiver operator characteristic (ROC) analysis confirming the relation between NLR and Tacrolimus overdose as presented in Figure 3. The ROC analysis for NLR above 3.62 shows predictive properties for Tacrolimus overdose (over 15 ng/mL) (AUC = 0.633, *p* = 0.008) yielding a sensitivity of 50% and specificity of 83.73%.

The ROC analysis for MPV above 9.9 revealed predictive properties for Tacrolimus overdose (over 15 ng/mL) (AUC = 0.527, *p* = 0.564) yielding sensitivity 22.92% and specificity 91.27%.

## 5. Discussion

The tacrolimus is a crucial component of antirejection therapy after heart transplantation due to superior results in outcomes, as compared to cyclosporine [16]. Immunosuppressant drugs may cause many side effects, including kidney dysfunction, hypertension, post-transplant diabetes mellitus, increased infection risk, and hyperlipidemia, which highlights the importance of individualized therapy [17].

Triple therapy, including calcineurin inhibitor (CNI) and mycophenolate mofetil with corticosteroids, is still the gold antirejection therapy standard. Interactions between aforementioned drugs were postulated. The coadministration of calcineurin inhibitors with mycophenolate mophetil (a prodrug) may be influenced by its enterohepatic recirculation, which allows the former to be converted into an active drug [18]. These factors illustrate the significance of proper dosages of drugs that are monitored by serum levels in specialized centers. These factors are among the most crucial, since they may influence drug serum levels and cause difficulties in proper drug dosing. Additionally, that serum level monitoring is performed only in highly specialized centers. Personalized monitoring is important to control drug dosages and minimize the risk of overdose and drug toxicity.

Variability in serum calcineurin inhibitor levels were presented by Verdu et al. [19]. Drug overdose was related to acute kidney injury in multivariable analyses performed in 107 patients by Pembele et al. [20].

Our study was focused on simple whole blood parameters which could indicate drug overdose and suggest the necessity for more detailed testing in reference centers. Since immunosuppression therapy after heart transplantation in based on calcineurin inhibitor (CNI based therapy), we believed that close monitoring would prove crucial for efficient treatment of heart transplant recipients. We found a relationship between CNI overdose and NLR that could be easily monitored, reducing the risk for side effect occurrence in populations of heart transplant recipients. To our best knowledge, this was the first study to present a relationship between simple hematologic indices and tacrolimus serum levels that could enable personalized therapy after heart transplantation.

Neutrophil-to-lymphocyte ratio is a simple parameter, obtained from whole blood count, that describes inflammatory states with possible clinical significance [21,22]. Its significant values for risk of atherosclerosis progression were previously reported [23,24,25].

Immunosuppressive agents applied in solid organ transplantation significantly reduce rejection rates but fail to control inflammatory reactions related to innate immunity [26]. The novelty of our study lay in the presentation of evidence showing that tacrolimus overdose may be related to NLR increase as shown in innate immunological response overreaction. The results of Shao et al. revealed that calcineurin inhibitor FK506 (tacrolimus) controlled allograft immunity directly through control of T cell immunity, independent of myeloid innate immune cells. The results gave a new perspective on innate immunity control, at least with the utility of such a simple marker as NLR. Moreover, the increase of NLR in our study, correlated with tacrolimus overdose, may indicate innate immunology response activation, while immunosuppressants accumulate in excess doses.

Our study not only pointed out the relationship between tacrolimus overdose and NLR increase, but may also indicate innate immunity reactions due to drug accumulation.

The second parameter that we took into consideration was mean platelet volume (MPV). Platelets play an important role as mediators of inflammation, including their interaction with leucocytes [15]. Therefore, platelet indices have been investigated in clinical conditions related to inflammatory response. The MPV prognostic value in chronic lymphocytic leukemia was shown by Masternak et al. [27]. Cardiovascular risk prognosis related to MPV values was postulated by Crafa et al. [28]. Hepatocellular carcinoma recurrence after liver transplantation predicted by MPV was proven by Zhang et al. [29]. In our study, the higher MPV was related to higher tacrolimus serum levels. While we presented higher NLR reflecting inflammatory response related to tacrolimus overdose, higher MPV may result from the aforementioned platelet-to-leucocytes interaction. According to the meta-analysis conducted by Tullavardana et al., mean platelet volume (MVP) was a potential biomarker for the diagnosis of acute states, including appendicitis [30]. Meyer et al., in their study, presented inflammatory cascade induction via NF-κB9 (nuclear factor kappa-light-chain-enhancer of activated B cells), signaling endothelial colony progenitor cells induced by tacrolimus [31].

We believe that the results of our study, despite many limitations, may point to a new perspective for heart recipient monitoring, especially in outpatient clinics in which measuring immunosuppressive drug serum levels is not feasible. Simple blood parameters may help clinicians to detect a risk of tacrolimus overdose and take further steps in certain situations. This is especially true for long-term follow ups, when contact with reference hospitals is infrequent.

### Study Limitation

This was a single-center retrospective study, and can be regarded as the initial report. As such, further studies are required.

## 6. Conclusions

Our results suggested that NLR above 3.62 could be regarded as a simple indicator of tacrolimus overdose in an outpatient clinic.

## Figures and Tables

**Figure 1 diagnostics-12-00037-f001:**
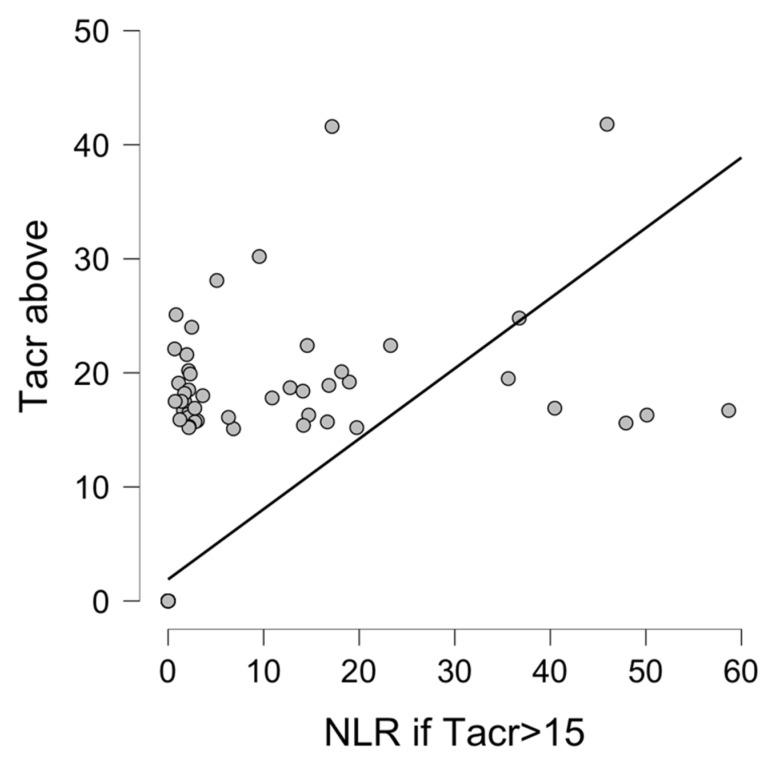
NLR if Tacrolimus C-0 serum level above 15 ng/mL. Correlation between NLR and Tacrolimus overdose (Tacr serum level > 15 ng/mL). Abbreviations: NLR—neutrophil-to-lymphocyte ratio, Tacr—tacrolimus.

**Figure 2 diagnostics-12-00037-f002:**
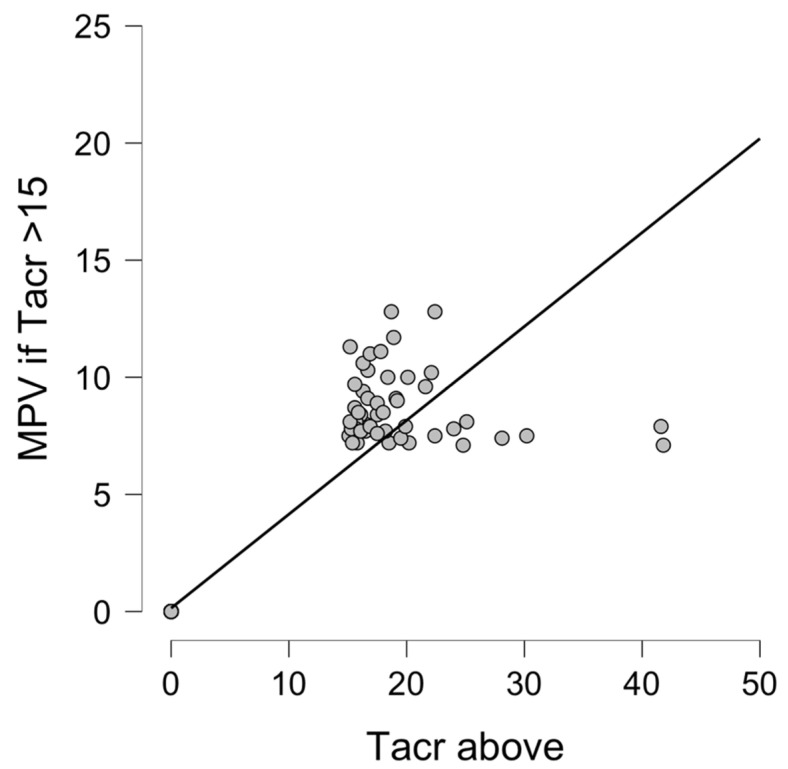
Correlation between MPV and Tacrolimus overdose (Tacr serum level > 15 ng/mL). Abbreviations: MPV—mean platelet volume, Tacr—tacrolimus.

**Figure 3 diagnostics-12-00037-f003:**
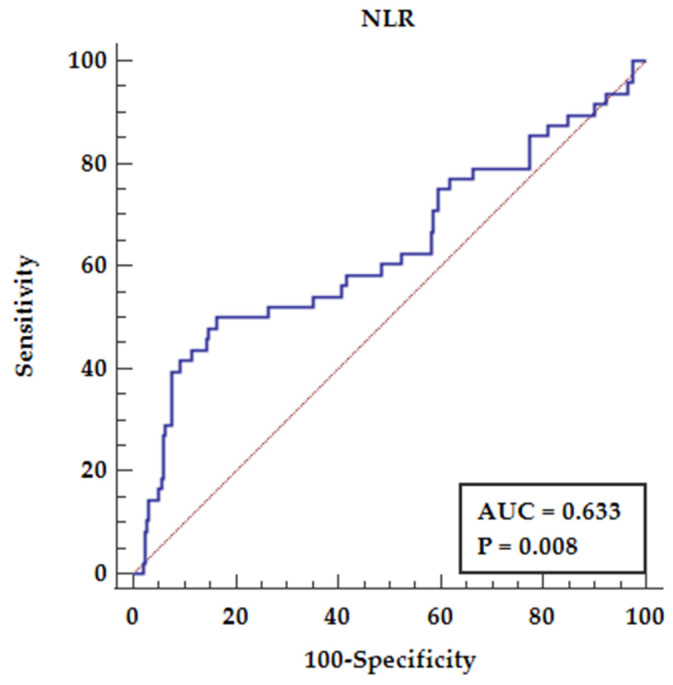
Receiver operator characteristics for NLR and tacrolimus overdose (>15 ng/mL). The ROC curve analysis of the NLR for predicting tacrolimus overdose. Abbreviations: NLR—neutrophil to lymphocyte ratio.

**Table 1 diagnostics-12-00037-t001:** Demographical and clinical data.

Parameters.	No = 30
Age (years):	43 ± 13
Gender (Male/Female):	23 (77%)/7 (23%)
Indication for transplantation:	
1. DCM	17 (57%)
2. ICM	12 (40%)
3. HCM	1 (3%)
Cold ischemia time (minutes):	218 ± 35
Concomitant diseases:	
1. arterial hypertension (%)	6 (20%)
2. diabetes. mellitus (%)	3 (10%)
3. kidney dysfunction (%)	2
4. hypercholesterolemia (%)	3 (10%)
Postoperative hospitalization (days):	28 ± 12
Rejection episodes:	
1.within 1st postoperative year	2 (10%)
2. during study period (beyound 12 months after transplantation)	0 (0%)
Immunosuppression protocol:	
1. Tacr + MMF + GS	28 (90%)
2. Tacr + MMF	2 (10%)

Abbreviations: DCM—dilated cardiomyopathy, ICM—ischemic cardiomyopathy, GS—glicocorticosteroids, HCM—hyperthrophic cardiomyopathy, MMF—mycophenolate mophetil, Tacr—tacrolimus.

**Table 2 diagnostics-12-00037-t002:** Immunosuppressive drug serum levels and NLR and MPV throughout the observation period.

Intervals	Tacrolimus (Median; Q1–Q3)	NLR (Median; Q1–Q3)	MPV (Median; Q1–Q3)
Time after heart transplantation:			
• 14 months (median; Q1–Q3)	9.7 (8.4–13.3)	2.11 (1.60–3.39)	8.7 (7.7–9.1)
• 16 months (median; Q1–Q3)	9.3 (7.4–13)	2.17 (1.61–3.39)	8.6 (7.7–9.1)
• 18 months (median; Q1–Q3)	9.8 (7.5–12.2)	2.70 (2.16–3.71)	8.8 (7.9–9.4)
• 20 months (median; Q1–Q3)	9.1 (7.5–12.2)	2.57 (1.99–5.04)	8.6 (7.8–9.1)
• 22 months (median; Q1–Q3)	10.8 (7.7–12.8)	2.57 (2.04–3.83)	8.5 (7.6–9.1)
• 24 months (median; Q1–Q3)	10.1 (8.6–13.3)	2.27 (1.90–3.93)	7.9 (7.4–8.9)
• 26 months (median; Q1–Q3)	10.1 (7.6–15.1)	2.46(1.74–3.91)	8 (7.4–9.3)
• 28 months (median; Q1–Q3)	9.1 (7.9–13.3)	2.31 (1.92–4.46)	7.9 (7.3–8.9)
• 30 months (median; Q1–Q3)	10.7 (8.4–15.5)	2.69 (1.64–6.83)	7.8 (7.4–8.2)

Abbreviations: MPV—mean platelet volume, NLR—neutrophil-to-lymphocyte ratio.

## Data Availability

Data supporting reported results can be found by direct contact with corresponding author after justifiable explanation of requirements for 3 years following the publication.

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
