# Peer review of "Neutrophil to Lymphocyte Ratio (NLR) as an Easily Accessible Parameter for Monitoring Tacrolimus Overdose after Heart Transplantation—Experimental Study"

_diagnostics, 2021, doi:10.3390/diagnostics12010037_

Round 1

Reviewer 1 Report

Congratulations for this interesting article. An article of importance in its field.
The studied topic presents practical importance. It's a good article. It could be an idea to continue the data collecting and extend the study to other hospital. 

The paper is well written and correct from a methodological point of view. 

Author Response

                                               Poznan, December 18th, 2021

Dear Reviewer,

Thank you for your valuable comments.

We feel privilege to be given you opinion.

Kind regards,

Tomasz Urbanowicz

Reviewer 2 Report

the aim of this study is to find a new simple hematological indices to help clinicians in immunosoppressive therapy monitoring. the authors identified the NRL indice as potential tool for monitoring drug overload. Actually, the effectiveness of patient monitoring is an important point. Despite the paper has some potential, there are a number of issue which should be adressed before the manuscript can be accepted for pubblication, as detail below:

1- NRL is a blood test that monitors infection, so authors should be better frame this index in solid organs transplantation

2- the authors should be describe better materials and methods

3- number of patients is not big but adding some data and analysis the manuscript could be improve in its clinical significance: rejection score of each patientes, NRL in the first year from transplantation and after (for example) a change of therapy.

4- provide rejection score of each patients, correlation between Rejection score with NRL

5- some data from blood into 1 year from heart tranplant? could be great have some data.

6- provide a more complete statistical analysis with some more different tests such as machine learning tests

7- RESULT SECTION: please provide a better description. Try to lead the reader by hand to the conclusions.

8- some data need to be explained. line 113 " analysis for NRL above 4.9" ?

9- the same at line 116 of the results

10- Figure legends and captions need to be improved adn explained

11- the discussion should be improved

Author Response

Poznan, December 18th, 2021

Dear Reviewer,

Thank you for your valuable comments. We corrected the manuscript according to them.

As follows:

1- NRL is a blood test that monitors infection, so authors should be better frame this index in solid organs transplantation

Dear Reviewer, thank you for your suggestion. Best to our knowledge NLR is an established marker of inflammatory reactions (not for infections by itself only) and has gained a wide interest due to its’ value in as a prognostic marker of systemic inflammation in several cardiovascular diseases, including heart failure, aortic stenosis, acute myocardial infarction, and outcomes after coronary revascularization (see below)

Urbanowicz TK, Olasińska-Wiśniewska A, Michalak M, Straburzyńska-Migaj E, Jemielity M. Neutrophil to lymphocyte ratio as noninvasive predictor of pulmonary vascular resistance increase in congestive heart failure patients: Single-center preliminary report. Adv Clin Exp Med. 2020 Nov;29(11):1313-1317. doi: 10.17219/acem/126292. PMID: 33269817.

Azab B, Zaher M, Weiserbs KF et al. Usefulness of neutrophil to lymphocyte ratio in predicting short- and long-term mortality after non-ST-elevation myocardial infarction. Am J Cardiol. 2010;106(4):470-476. doi: 10.1016/j.amjcard.2010.03.062.

Khalil C, Pham M, Sawant AC, Sinibaldi E, Bhardwaj A, Ramanan T, Qureshi R, Khan S, Ibrahim A, Gowda SN, Pomakov A, Sadawarte P, Lahoti A, Hansen R, Baldo S, Colern G, Pershad A, Iyer V. Neutrophil-to-lymphocyte ratio predicts heart failure readmissions and outcomes in patients undergoing transcatheter aortic valve replacement. Indian Heart J. 2018 Dec;70 Suppl 3(Suppl 3):S313-S318. doi: 10.1016/j.ihj.2018.08.002. Epub 2018 Aug 17. PMID: 30595282; PMCID: PMC6310731.

Angkananard T, Anothaisintawee T, McEvoy M, Attia J, Thakkinstian A. Neutrophil Lymphocyte Ratio and Cardiovascular Disease Risk: A Systematic Review and Meta-Analysis. Biomed Res Int. 2018 Nov 11;2018:2703518. doi: 10.1155/2018/2703518. PMID: 30534554; PMCID: PMC6252240.

Urbanowicz TK, Michalak M, Gąsecka A, Olasińska-Wiśniewska A, Perek B, Rodzki M, Bociański M, Jemielity M. A Risk Score for Predicting Long-Term Mortality Following Off-Pump Coronary Artery Bypass Grafting. J Clin Med. 2021 Jul 7;10(14):3032. doi: 10.3390/jcm10143032. PMID: 34300198; PMCID: PMC8305554.

Olasińska-Wiśniewska A, Perek B, Grygier M, Urbanowicz T, Misterski M, Puślecki M, Stefaniak S, Stelmark K, Lesiak M, Jemielity M. Increased neutrophil-to-lymphocyte ratio is associated with higher incidence of acute kidney injury and worse survival after transcatheter aortic valve implantation. Cardiol J. 2021 Nov 17. doi: 10.5603/CJ.a2021.0149. Epub ahead of print. PMID: 34787890.

2- the authors should be describe better materials and methods

            Thank you for this remark, we corrected that part accordingly.

3- number of patients is not big but adding some data and analysis the manuscript could be improve in its clinical significance: rejection score of each patientes, NRL in the first year from transplantation and after (for example) a change of therapy.

& 4- provide rejection score of each patients, correlation between Rejection score with NRL

Dear Reviewer thank you for your suggestion.

Several rejection scores have been proposed in the literature, however the common feature of them is the fact that the score shall be calculated at the transplant time to predict one-year outcome. Generally, the score is not routinely applied in everyday practice, while our aim was to assess simple, easily available, and repetitive parameter for evaluation of tacrolimus overdose. The rejection score, once calculated, is constant and does not change throughout the follow-up. In our paper, not a rejection risk but a tacrolimus overdose during follow-up after HTX was the point of interest. Moreover, we did not observe rejections in the study population.

5- some data from blood into 1 year from heart tranplant? could be great have some data.

Dear Reviewer thank you for your suggestion. However, our study period beginning has been established at the 12. month after HTX and subsequently patients were followed patients for 18 months until the 30th post-transplant month (this is described in Methods section and in Table 2). During the whole study period (18 months) the same parameters were collected and the analysis was focused on immunosuppression (tacrolimus) serum levels.

6- provide a more complete statistical analysis with some more different tests such as machine learning tests

Dear Reviewer thank you for your suggestion, a profound statistical analysis has been done by professional statistician, the co-author of the study and the paper (Prof Michal Michalak from Department of Computer Science and Statistics).

Machine learning tests require hundreds and thousands of data. Therefore, our study is not relevant for such an analysis due to two causes – first, the aim was to find a simple and easily available marker of tacrolimus overdose in a daily practice, and second we evaluated 30 patients with hundreds of NLR values.

7- RESULT SECTION: please provide a better description. Try to lead the reader by hand to the conclusions.

Dear reviewer, thank you for your suggestion, we corrected the results section and added the comment in the Discussion:

Since immunosuppression therapy after heart transplantation in based on calcineurin inhibitor (CNI based therapy) we believe that close monitoring is crucial for efficient treatment of heart transplant recipients.  

8- some data need to be explained. line 113 " analysis for NRL above 4.9" ?

& 9- the same at line 116 of the results

Dear Reviewer thank you for your suggestion, we corrected the information regarding ROC analysis. The cut off point for Tacrolimus overdose was 3,62 not 4,9. Thank you for your valuable suggestion.

We also performed receiver operator characteristic (ROC) analysis confirming the relation between NLR and Tacrolimus overdose as presented in Figure 3. The ROC analysis for NLR above 3,62 4.9 shows predictive properties for Tacrolimus overdose (over 15 ng/ml) (AUC =0.633, p=0.008) giving sensitivity 50% and specificity 83.73%.

10- Figure legends and captions need to be improved adn explained

We corrected the figure legend and captions

11- the discussion should be improved

We corrected the discussion section.

Kind regards,

Tomasz Urbanowicz

Reviewer 3 Report

The authors have described the increase in NLR with Tacro overdose. This is a simple blood test which can help manage Tacro dose in heart transplant recipients

  1. Methods: Authors should describe what is the recommended tacro levels.
  2. Authors should describe how MPV is estimated.
  3. How was Myocophenolate dose monitored?
  4. Is there any correlation between Cellcept level and NLR or MPV
  5. in 90% cases patients are on GC, as it is known to cause increase in neutrophil count. As it is excepted that Tacro and Cellcept can decrease lymphocytes, so the neutrophil count increases and the NLR also increase. The correlation of Tacro with innate immunity stimulation is over extrapolation of their results.

Author Response

Poznan, December 18th, 2021

Dear Reviewer,

Thank you for your valuable comments. We corrected the manuscript according to them.

As follows:

  1. Methods: Authors should describe what is the recommended tacro levels.

Dear Reviewer, thank you for your suggestion. We added:

The tacrolimus serum C-0 dose recommended for the analyzed treatment period is within range of 5-15 ng/ml.

  1. Authors should describe how MPV is estimated.

Dear Reviewer, thank you for your suggestion. The mean platelet volume (MPV) was presented as measurement of the dimension of platelets, calculated by hematological analyzers depending on the volume distribution during routine blood morphology test in Central Clinical certified Laboratory of Poznań University of Medical Sciences.

  1. How was Myocophenolate dose monitored?

Dear Reviewer, thank you for your suggestion. We added:

The separate measurements within 18 months’ time were collected combining 300 tests. The tacrolimus (FK506) and mycophenolate mophetil serum levels were collected as C-0 levels ( serum concentration beforemorning dose of drugs administration). 

  1. Is there any correlation between Cellcept level and NLR or MPV

Dear Reviewer, thank you for your suggestion. The mycophenolate mophetil serum levels were within therapeutic range, we did not find the correlation. Therefore, the results can’t be regarded as significant as there were neither overdose nor underdose events registered.

  1. in 90% cases patients are on GC, as it is known to cause increase in neutrophil count. As it is excepted that Tacro and Cellcept can decrease lymphocytes, so the neutrophil count increases and the NLR also increase. The correlation of Tacro with innate immunity stimulation is over extrapolation of their results.

Dear Reviewer, thank you for your suggestion. The study was performed 12 months past heart transplantation. The GC dose was 5-10mg per patients, therefore the effect on the neutrophils count was not exaggerated too much. However, we believe that new research is worth to be launched for patients monitoring in the early post-transplant period to collect the data from patients treated with wide range of GC doses.

We added:

During the study time patients were on low doses of steroids therapy including 5 +/- 2.5mg of prednisolone daily and mycophenolate mophetil serum C-0 levels were estimated as 2.9 +/- 0.3 mcg/ml.

                        NLR is calculated from neutrophils and lymphocytes count. Neutrophiles represent innate immunity therefore we believe that our results may show the relationship between innate immunity changes and tacrolimus overdose.

Kind regards,

Tomasz Urbanowicz

Round 2

Reviewer 2 Report

Results are still difficult to understand and figure legends without results are difficult to read.

Reviewer 3 Report

Congratulations. I have no further querries.